



# The rate and extent of windgap migration regulated by tributary confluences and avulsions

Eitan Shelef[1] and Liran Goren[2]

[1]Department of Geology and Environmental Science, University of Pittsburgh, Pittsburgh, PA, 15217
[2]Department of Earth and Environmental Sciences, Ben-Gurion University of the Negev, Israel, 84105

**Correspondence:** Eitan Shelef (shelef@pitt.edu)

**Abstract.** The location of drainage divides sets the distribution of discharge, erosion, and sediment flux between neighboring basins, and may shift through time in response to changing tectonic and climatic conditions. Major divides commonly coincide with ridgelines, where the drainage area is small and increases gradually downstream. In such settings, divide migration is attributed to slope imbalance across the divide that induces erosion rate gradients. However, in tectonically affected region, low-relief divides, windgaps, abound in elongated valleys, whose drainage area distribution is set by the topology of large, potentially avulsing side-tributaries. In this geometry, distinct dynamics and rate of along-valley windgap migration is expected, but this process remains largely unexplored. Inspired by field observations, we investigate along-valley windgap migration by simulating the evolution of synthetic and natural landscapes, and show that confluences with large side tributaries influence migration rate and extent. Such confluences facilitate stable windgap locations that deviate from intuitive expectations based on symmetry considerations. Avulsions of side tributaries can perturb stable windgap positions and avulsion frequency governs the velocity of windgap migration. Overall, our results suggest that tributaries and their avulsions may play a critical role in setting the rate and extent of windgap migration along valleys and thus the time scale of landscape adjustment to tectonic or climatic changes across some of the most tectonically affected regions of Earth, where windgaps are common.

## 1 Introduction

Drainage divides play a pivotal role in controlling the geometry of fluvial landscapes and shaping their hydrologic and geomorphic functionality. Divides' location, and as a consequence, basins' geometry, are transient features of the landscape (e.g., *Willett et al.*, 2014; *Whipple et al.*, 2017) that respond to shifting boundary conditions. As early as the late 19th century, *Gilbert* (1877) and *Davis* (1889) described divide migration and drainage reorganization, and since then, the geomorphology literature has documented many examples of such drainage network reorganization at both the local-scale (e.g., *Johnson*, 1907; *Woodruff*, 1977; *Nugent*, 1990; *Brocard et al.*, 2011; *Prince et al.*, 2011; *Yanites et al.*, 2013; *Schmidt et al.*, 2015; *Forte et al.*, 2015; *Fan et al.*, 2018) and the regional-scale (e.g., *Ollier*, 1995; *Zelilidis*, 2000; *Shephard et al.*, 2010; *Clark et al.*, 2004; *Liu*, 2014; *Yang et al.*, 2020).

Whereas drainage divides could be breached abruptly via river capture events (e.g., *Bishop*, 1995; *Prince et al.*, 2010; *Willett et al.*, 2014), a possibly more common process involves long-lasting and continuous divide migration at basins headwaters,





where the divides are located along ridge lines (e.g., *Willett et al.*, 2014; *Goren et al.*, 2014b; *Shelef and Hilley*, 2014; *Whipple et al.*, 2017; *Beeson et al.*, 2017; *Braun*, 2017). In these high relief settings, divide migration is linked to an imbalance in erosion rate across the hillslopes that bound the divide. Hillslope erosion and slope are linked to incision at the proximal channel head (the local base level for the hillslope). Given that channel erosion rate scales with channel gradient and drainage area (a proxy for discharge, (e.g., *Howard*, 1994; *Whipple and Tucker*, 1999)), small across-divide differences in these factors can lead to

disparate channel incision rates across the divide and to a gradual divide migration directed from the fast to the slow eroding hillslope (*Beeson et al.*, 2017; *Braun*, 2017).

Over long timescales, feedbacks might arise that promote a prolonged and gradually declining divide migration (*Willett et al.*, 2014; *Whipple et al.*, 2017). An area-feedback occurs as the basin at the side of the fast eroding hillslope grows (hereafter the aggressor basin), whereas its across-divide neighbor shrinks (hereafter the victim basin) (*Willett et al.*, 2014; *Yang et al.*, 2015;

*Whipple et al.*, 2017). An increase in the aggressor's drainage area, increases its erosion rate and promotes further divide migration and area gain (*Mudd and Furbish*, 2005; *Whipple et al.*, 2017; *Goren et al.*, 2014b). The opposite process operates at the victim basin that losses drainage area. Parallel to this area-feedback, which furthers divide migration, a channel length-feedback arises which gradually suppresses this migration. As the divide migrates, the victim basin shortens and the aggressor basin lengthen, which increases the overall gradient of the victim basin compared to the aggressor basin, eventually leading to

a balance in erosion rate across the divide so the divide migration stops. The relative magnitude of these competing area and length feedbacks changes gradually through the migration process, and thus the velocity of divide migration typically declines smoothly through time (*Braun*, 2017). This gradual divide migration process is associated with settings whereby the migrating divide is located along a ridgeline that is topographically higher than the tributaries draining to the victim basin. Migration then erases the morphology and topology of the victim basin and obliterates the antecedent course of victim basin's tributaries

(Figure 1a).

A distinctly different dynamic is expected to emerge when a drainage divide forms a deep saddle within a valley (a windgap, (e.g., *Bishop*, 1995)). In such settings, the windgap is lower than the ridges that bound the valley, and thus the morphology of the bounding ridge lines and the tributaries that drain them into the valley can be preserved despite windgap migration along the valley (e.g., *Harel et al.*, 2019) (Figure 1b). The preserved side tributaries can cause punctuated changes in drainage area

through the divide migration process that distinctively differ from the aforementioned gradual exchange of drainage area. We thus intuit that the coupled dynamics of side tributaries draining close to a windgap and windgap migration could be key in controlling windgap stability, migration velocity and the evolution of valley topography.





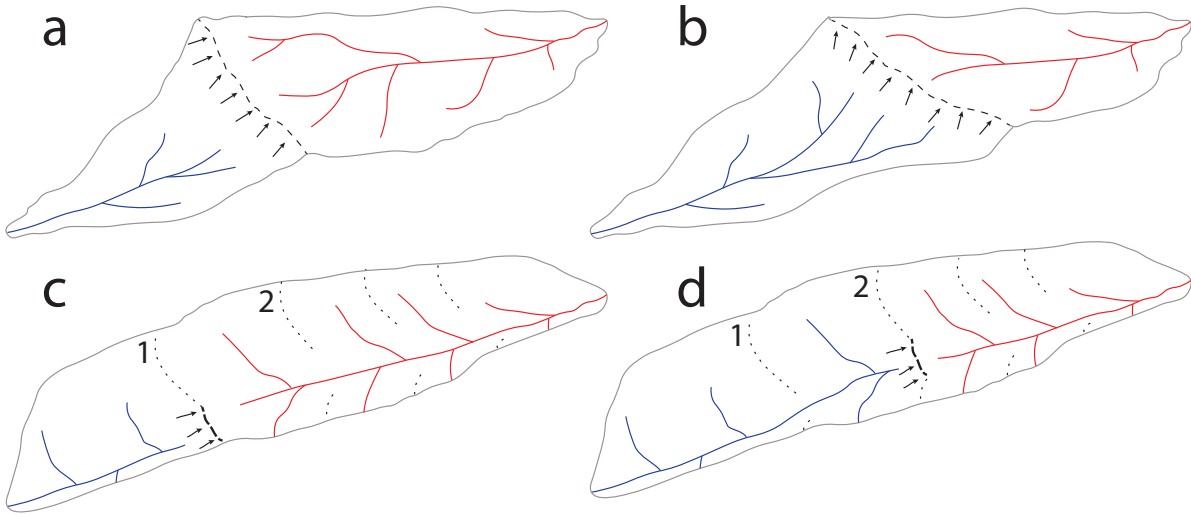

**Figure 1.** Schematic illustration of a typical (a,b) vs. along valley (c,d) divide migration. (a) Illustration of a typical divide migration process where the divide (dashed line) is located along a ridge line (after *Whipple et al.* (2017)). (b) The same setting as in panel a, after the divide migrated some distance. Note that as the ridge line migrates it erases the tributaries of the victim basin. (c) Illustration of a windgap (dashed line) migration along a valley. Note that the tributaries that drain to the valley can be preserved through the migration process, so that the migrating windgap can traverse confluences with tributaries through its migration. (d) Same setting as in panel c, after the windgap migrated some distance. Note that the low order drainage divides between tributaries (dotted lines) can merge with the migrating windgap to form a high order divide (for example see the low order divides marked 1 and 2 in panels c and d).

Windgap migration along a valley is likely common in tectonically active and/or structurally deformed areas where windgaps are prevalent. Such migration is likely facilitated by relatively erodible bedrock or sediments within valleys (*Harel et al.*, 2019), or may be induced by tilting in tectonically active areas (*Bishop*, 1995; *Clark et al.*, 2004). Windgaps are found along longitudinal, structurally controlled valleys as well as in antecedent highland valleys truncated by cliffs or formed by large capture events (e.g., *Haworth and Ollier*, 1992; *Bishop*, 1995; *Prince et al.*, 2010; *Harel et al.*, 2019) (e.g., Figure 2). The location and migration of windgaps dictate the distribution of erosion, discharge, and sediment fluxes between diverging regional to continental scale drainage networks and sedimentary basins, and may therefore set a primary control on the geologic evolution of some of the most active regions on Earth. Yet, the dynamics of windgap migration remain largely unexplored.

In this study we set to identify and explore key aspects of the dynamics of windgap migration. In particular, we focus on the influence of confluences and avulsions of side tributaries on the rate and extent of windgap migration.




Earth **Surface**
Dynamics
Discussions

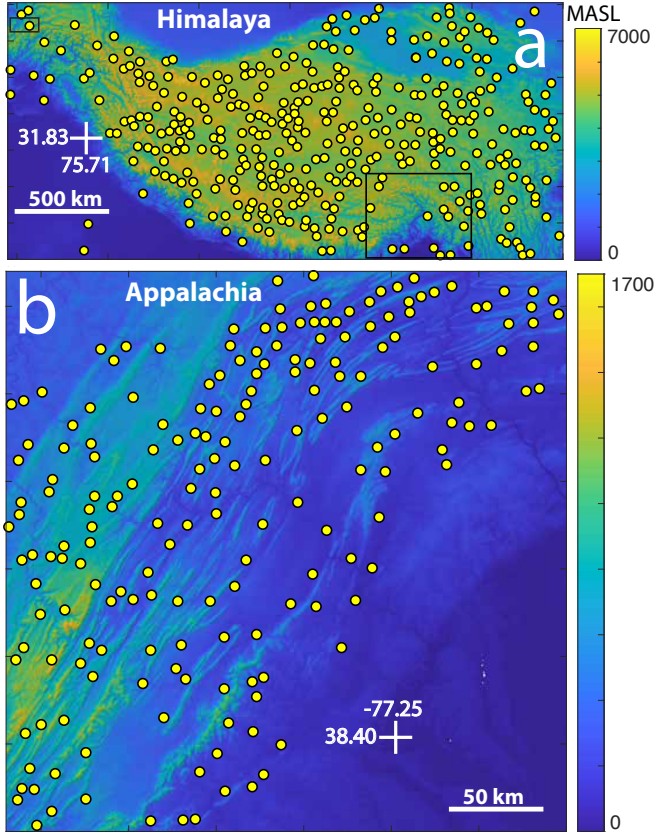

**Figure 2.** Windgaps in tectonically influenced areas. (a) A map of estimated windgap locations (yellow circles) in the Himalaya. The inset box at the bottom right marks the areas shown in figures 3c and 6, and the box at the upper left marks the area shown in Figure 4d-e. (b) A map of estimated windgap locations in the Appalachian fold-thrust belt. Windgaps locations were identified over GMTED DEMs (*Danielson and Gesch*, 2011) by delineating the drainage divides associated with a channel network mapped with drainage area thresholds of 200 and 12.5 km$^2$ for the Himalayas and Appalachia, respectively (values hereafter are reported in the same order for these two areas). To identify major divides at relatively low topographic positions (i.e., windgaps) we isolated high order divides (*Scherler and Schwanghart*, 2020), computed the local relief based on divide elevations within a radius of 30 and 5 km, respectively, and identified locations of minimal divide elevation within these radii. These radii are based on typical valley widths in these areas and are similar to the typical distance from divide to channel head based on the drainage area threshold that was used to define channels (scaled to distance via Hack's law). Of the locations of minimal divide elevation, we identified windgaps as locations with low along valley relief and high across valley relief, to do so, we identified locations of local divide relief that is higher than 200 and 100 m, respectively (i.e., common low-end reliefs for noticeable valleys in these areas), and a vertical elevation difference between the divide and closest streams (on each side of the divide) that is less than 20% of the local divide relief.





## 2 Field observations

This study is inspired by field observations of windgaps that traversed confluences with side tributaries along their migration

pathway. Along the Arava escarpment, Israel, antecedent valley systems were beheaded during a regional drainage reorganization associated with the escarpment's development (*Ginat et al.*, 2000; *Avni et al.*, 2000; *Harel et al.*, 2019), forming numerous windgaps that are aligned with the escarpment cliff. Some of the windgaps migrated inland along antecedent valleys (*Harel et al.*, 2019), traversing confluences with side tributaries. Observations from Wadi Grofit, for example, show that several confluences were traversed by a migrating windgap, as seen by their barbed morphology (Figure 3a-b). A similar setting, albeit at a

much larger scale, is observed at the eastern syntaxis of the Tibetan Plateau. Here, the Parlung-Siang-Lohit river capture (*Lang and Huntington*, 2014; *Schmidt et al.*, 2015; *Govin et al.*, 2018; *Zhang et al.*, 2019) (Figure 3c) triggered a windgap migration of more than 200 km along the Parlung valley. Through its migration, the windgap traversed confluences with side tributaries that may have influenced the migration dynamics.

This study is further motivated by field observations showing that avulsions of side tributaries can shift discharge across

windgaps. Such a setting is observed, for example, along an east-west directed valley, next to Mt. Berech on the highlands of the Arava escarpment. Here, avulsions occur at the head of an alluvial fan that is formed at the mouth of a side tributary that drains into a valley close to a windgap. The avulsions route a fraction of the discharge of the side tributary to the escarpment side of a windgap (Figure 4a-c), where the magnitude of incision appears to be comparably high. A similar example, of a somewhat larger scale, is observed in the Hindu-Kush province of the Himalaya, next to the Ishkashim Pass windgap, Afghanistan. Here,

a side tributary forms an alluvial fan as it drains into an east-west directed valley, and avulsions at the apex of this fan route discharge across a windgap (Figure 4d-e). We intuit that such avulsions can modify the relative erosion rates across windgaps and thus the rate and extent of windgap migration.

The observations described above (Figures 3 and 4), and the realization that windgap migration may be a prominant mechanism of landscape development in tectonically active and structurally deformed regions (Figure 2), inspired simulations that

explored how the dynamics of windgap migration are influenced by: (a) confluences with side tributaries, and (b) avulsions of such side tributaries across windgaps.





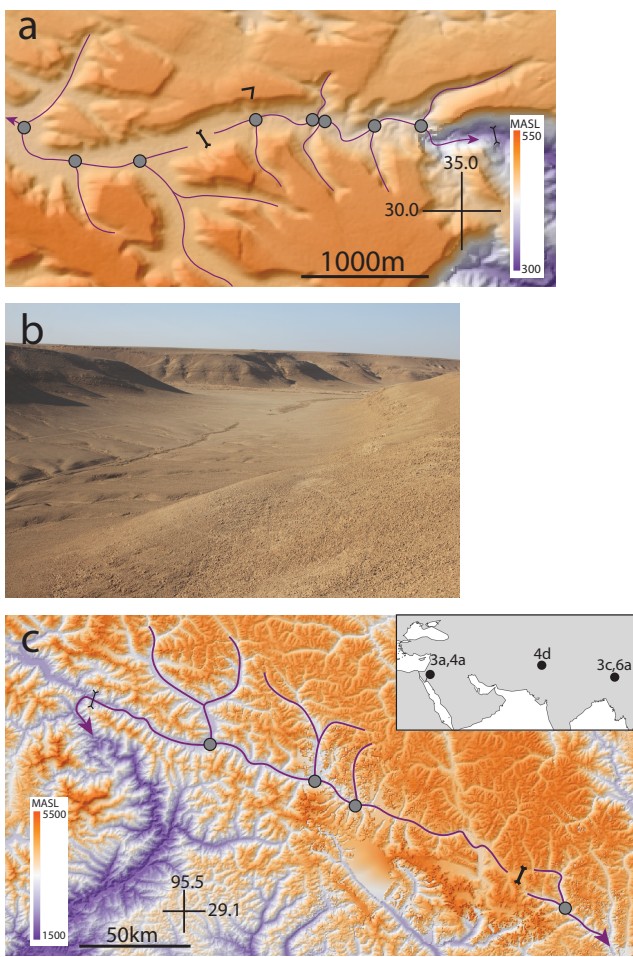

**Figure 3.** Examples of valleys with windgaps and major confluences with side tributaries (grey circles). Major channels are marked in blue and their flow direction is marked with an arrow. (a) A map, based on a TanDEM-X DEM, showing an example from wadi Grofit in the Negev highlands along the Arava escarpment, Israel. The current location of the windgap is marked by a bold forked black line. The approximated initial location of the windgap is marked with a thin forked black line (*Harel et al.*, 2019). A black v-shaped symbol open to the south-west shows the locations from which the pictures in panels b was taken. (b) A picture of the windgap shown in panel a, taken from the location of the aforementioned black v-shaped symbol towards south-west. (c) A map, based on GMTED2010 DEM, showing windgaps and confluences along the Parlung valley, China. The current location of the windgap is marked by a bold forked black line. A thin forked black line at the north-west portion of the map marks the approximate initial location of the windgap. In both the Grofit and Parlung exampels, the windgap likely traversed confluences with side tributaries (grey circles), resulting in their barbed morphology, as it migrated to its current location. The inset map shows the general location of the field examples presented in this and other figures, figure numbers are specified next to each location.





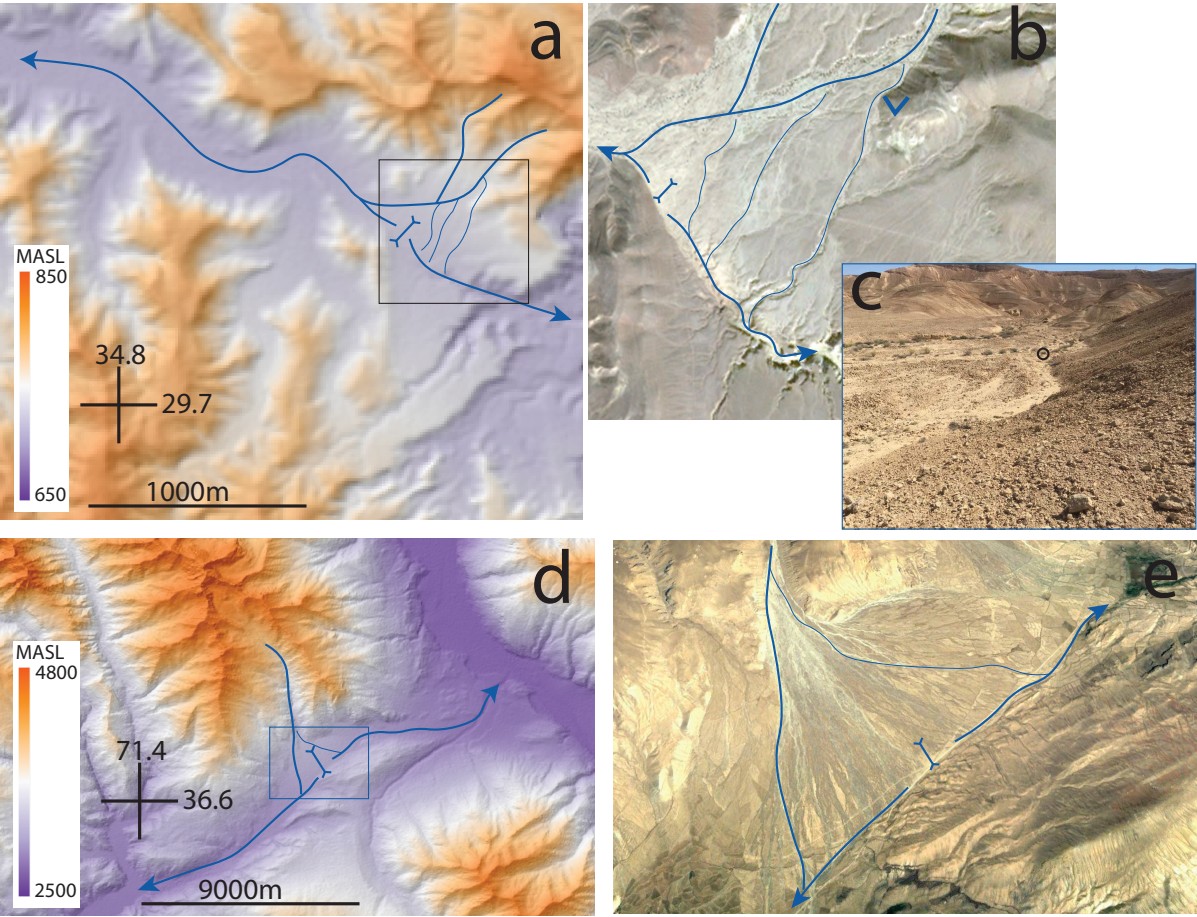

**Figure 4.** Examples of avulsions across windgaps (forked black line). Channels are marked in blue. (a) A map, based on a TanDEM-X DEM, of a windgap next to Mt. Berech in the Negev hilghlands along the Arava escarpment, Israel. Note that although the side tributaries marked in blue drain primarily north-west of the windgap, few bifurcating branches appear to route a fraction of the tributaries' discharge to the other side of the windgap. Black box marks the area shown in panel b. (b) An airphoto (©Google Earth 2020) showing the main channels of the side tributaries as well as their bifurcating branches. A black v-shaped symbol open to the north shows the locations from which the picture in panels c was taken. (c) View of an avulsion point looking upstream (north) from the aforementioned v-shaped symbol. Circle marks a backpack for scale, that is located at the bifurcation point. (d) A map, based on a GMTED2010 DEM, of the Ishkashim Pass area in Afghanistan. Note that although the side tributary (marked in blue) drains primarily south-west of the windgap, few bifurcating branches appear to route a fraction of its discharge to the other side of the windgap. Black box marks the area shown in panel e. (e) An airphoto (©Google Earth 2020) showing the bifurcation of a side tributary across the windgap.





## 3 Method

To investigate the influence of side tributaries on the velocity and extent of windgap migration, we use a landscape evolution model (e.g., *Tucker and Hancock*, 2010; *Perron et al.*, 2009):

$$\frac{dz}{dt} = U - (KA^m S^n - D\nabla^2 z). \tag{1}$$


In equation (1), $\frac{dz}{dt}$, the change in elevation, $z$ [L], through time, $t$ [T], is a function of uplift rate, $U$ [L/T], channel incision through detachment limited processes, $KA^m S^n$ [L/T], and changes in elevation due to diffusive sediment transport, $D\nabla^2 z$ [L/T] that likely dominates hillslope settings. In this model, channel incision is a function of drainage area, $A$ [L$^2$], and topographic gradient, $S$, as well as an erodibility coefficient, $K$ [L$^{1-2m}$ T$^{-1}$]. The two exponents, $m$ and $n$ acknowledge that

erosion may be a non-linear function of drainage area and gradient, respectively (*Howard and Kerby*, 1983; *Seidl and Dietrich*, 1992; *Howard*, 1994). Sediment transport is modelled as a diffusive process, where $D$ [L$^2$ T$^{-1}$] is a diffusion coefficient, and $\nabla^2 z$ [L$^{-1}$] is the laplacian of elevation (*Culling*, 1963; *Howard*, 1994). We use a finite difference scheme, where drainage area is the summation of the area of all upstream nodes, and topographic gradient is computed via a forward difference scheme in the downslope direction. The drainage area of the divide node is bifurcated between neighboring nodes according to the

relative magnitude of the slope to each neighboring node raised to a power of 1.1 (i.e., *Freeman*, 1991). This is a conservative choice that aims to minimize the influence of spatial discretization on windgap stability (i.e., *Pelletier*, 2004). The model integrates equation (1) through time using a 4-5 order explicit Runge-Kutta integration where time-stepping is constrained by the Courant criteria. The parameters $K$, $m$ and $n$ are determined based on common values published in the literature, D is scaled based on the values of $K$ and $m$ so that comparable models have the same length ($L_p$) associated with a Péclet value of

unity (i.e., $L_p = \left(\frac{D}{K}\right)^{\frac{1}{2m+1}}$ after *Perron et al.* (2008, 2009)). We set this length to be relatively short (200-500 meters) such that diffusive sediment transport is generally negligible within channels. The values of the parameters used in different simulations are reported in the captions of the figures that present these simulations.

To explore how confluences with side tributaries influences the migration velocity and stable positions of windgaps, we first simulate the evolution of a synthetic 1-dimensional landscape with such confluences (hereafter 'fixed confluence simulations').

The initial setting (Figure 5a,b) is of an elongate valley, where the windgap is at the left edge of the valley, bounded by a cliff to its left, and a trunk channel drains the valley to the right (Figure 5b, case 1). The initial topography of the trunk channel is set to be at a topographic steady state (i.e., $dz/dt = 0$) in accordance with equation (1). The boundary conditions are set to a constant and equal elevation at both ends of the model. The topology of the valley-tributary system is prescribed as a set of equally spaced trunk-tributary confluences along the valley, where tributaries have the same drainage area (i.e., the drainage

area added at the confluence) (Figure 5a) and the valley has a constant width. The deviation of this topology from a Hack scaling (i.e., *Hack*, 1957) is supported by field observations (i.e., Figures 3, 4) and also contributes to the simplicity of this synthetic model setting. We varied the tributaries drainage area between simulations and recorded the velocity of windgap migration and the location where the windgap attains a stable position. A stable position is defined as where the windgap position is fixed and the elevation difference between consecutive time steps is everywhere zero for 100 consecutive time steps.

We explored the influence of the scaling between erosion rate and drainage area by varying the value of the area exponent $m$



(0.45, 0.55). Simulation results are compared to a reference simulation, where there are no confluences with side tributaries, the local drainage area is identical for all model nodes, and the total drainage area is equal to the equivalent simulation with trunk-tributary confluences.

To study the influence of avulsions of side tributaries on the rate and extent of windgap migration, we simulated such avul-
sions by shifting the location of trunk tributary confluences through time (hereafter, 'avulsion simulations'). To achieve this dynamics, we randomly varied the confluence location within a prescribed distance from its initial position. The random distances are selected from a uniform distribution centered at the original location of each confluence, and the maximal distance is constrained to half the distance between the original location of confluences (i.e., Figure 5a). Avulsions occur in all tributaries, regardless of their location relative to the windgap. We explored the influence of the time span between avulsions (i.e., shifts
in confluence location) by varying it between simulations (from 50 to 500 yrs).

An independent set of simulations is dedicated to exploring the potential influence of tributary confluences on windgap migration in a natural setting. Here, the topology of the aforementioned Parlung-Siang-Lohit system is used as a template for the simulation, and we focus on the system dynamics following the Parlung-Siang-Lohit capture (*Lang and Huntington*, 2014; *Schmidt et al.*, 2015; *Govin et al.*, 2018; *Zhang et al.*, 2019) (Figure 6a-b). The initial conditions (Figure 6d) replicate
the inferred channel-system topography and topology at the time of the capture. For this 'natural' experiment, we assume that (a) the aggressor (Siang river) and victim (Parlung river) basins where approximately at a topographic steady state at the time of capture, (b) the capture occurred at point d in Figure 6a, (c) the channels' profile could be reconstructed based on equation (1), and (d) the location and drainage area of tributary confluences were similar to the present-day tributary confluence configuration. The confluence between the Siang and Lohit-Parlung (point f in Figure 6a) is used as the boundary conditions
for these simulations. We extract the drainage area along the rivers from a GMTED2010 DEM (*Danielson and Gesch*, 2011) with a resolution of 15 arc-seconds (approximately 500 m). We slightly modified the DEM to correct inaccuracies in basin boundaries close to the headwater of the Parlung river. The choice of model parameters (see caption of Figure 6) is coarsely guided by values suggested in the literature (*Wang et al.*, 2017; *Govin et al.*, 2018; *Yang et al.*, 2018; *Zhang et al.*, 2019) and adjusted to the present-day relief. Note that this simulation aims to demonstrate the potential influence of network topology on
windgap migration in a natural setting, and not to investigate the development of the Parlung-Siang-Lohit capture specifically.



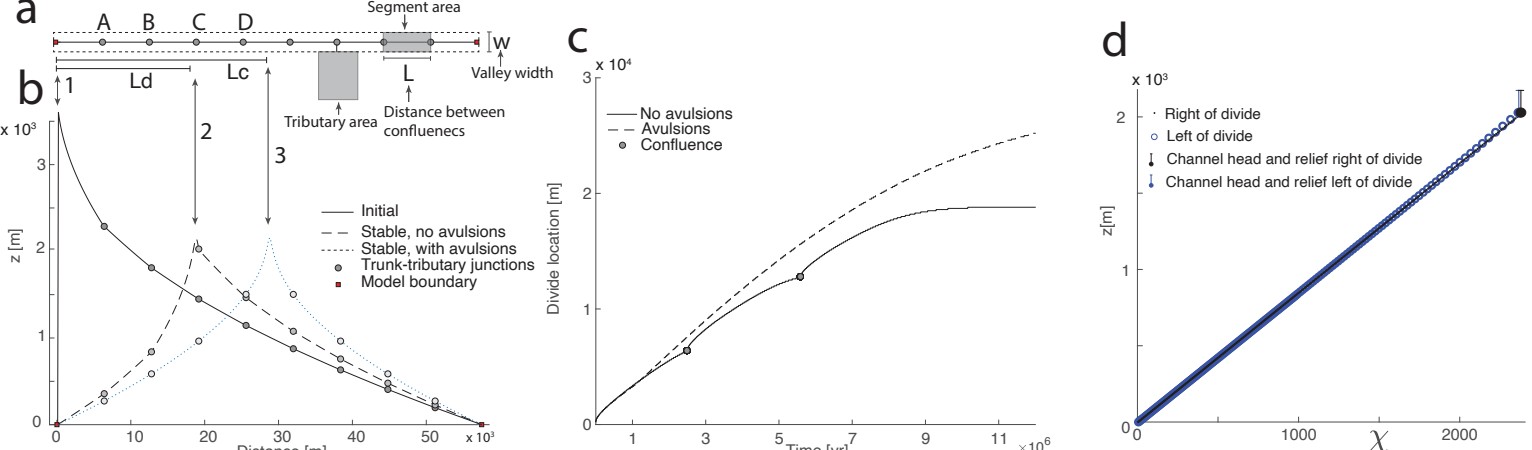

**Figure 5.** Simulations of along valley windgap migration across confluences with tributaries. (a) Plan view schematic of a 1-dimensional model domain that simulates a valley (valley boundaries marked by a dashed line) of constant width ($w$) that is drained by a trunk channel (dark line) with equally spaced ($L$) confluences with tributaries (grey circles). The drainage area of a segment between confluences is fixed ($L \times w$, schematically represented by a grey rectangle). The drainage area of a tributary is the local area that is added at each confluence (an example is represented here by a grey rectangle). $L_d$ and $L_c$ mark the distance from the left edge of the model domain to the location of a stable windgap position and to the center of the model domain, respectively, and are referred to in figures 7 and 8. Confluences marked A-D are referred to in figure 7a. (b) An example of simulated topographic profiles along the trunk channel: (1) A topographic profile of the simulation's initial condition (solid line, trunk-tributary confluences marked by dark grey circles); (2) Simulated steady state topography that develops from the initial condition in profile #1 through a fixed confluences simulation (dashed line, trunk-tributary confluences marked by medium grey circles). Note that the windgap attains a stable position away from the center of the model domain (i.e., $L_d < L_c$). Also note that this steady position occurs adjacent to a trunk-tributary confluence on the victim's side of the windgap; (3) Simulated steady state topography with avulsions (dotted line, light gray circles mark the mean location of trunk-tributary confluences - which is the same as that of the fixed confluences). (c) Simulated windgap location vs. time for the simulations in panel b. For the fixed confluences simulation (solid line), note the changes in windgap migration velocity (i.e., the gradient of the line in the figure) as the windgap migrates across tributary confluences (grey circles). Note that these changes in migration velocity are not apparent in a simulation with avulsions (dashed line) and that the overall windgap velocity is higher when avulsions are simulated. Note that the plot shows the model duration until the windgap in the fixed confluences simulation attained a stable position (case 2 in panel b), and that in the avulsions simulation the windgap continued migrating until it attained a stable position at the center of the model domain (case 3 in panel b, at a distance of about 30 km from the initial windgap location). (d) A $\chi - z$ plot (*Perron and Royden*, 2013) for case 2 in panel b. The plot demonstrates that this windgap position is stable although it is not in the center of the model domain. Note that the relief from each channel head to the windgap is also marked (see legend). The channel head is defined based on where the topographic profile shifts from concave to convex. The channel head right of the divide (a black filled circle in the $\chi - z$ plot) is at the adjacent tributary confluence (grey filled circle just right of the divide in profile 2, panel b). Panels a-d are based on a model configuration with 8 trunk-tributary confluences (4 at each side of the model center), 575 model nodes, node spacing: $\delta x = 100$, confluence spacing: $L = 6400$ m, valley width: $w = 400$ m, tributary area equals 2 segment areas (i.e., $2 * L \times w$), hillslope diffusion coefficient: $D = 0.24$ m$^2$ yr$^{-1}$, exponents: $n = 1$, $m = 0.45$, channel erodibility: $K = 1 \times 10^{-5}$ m$^{0.1}$ yr$^{-1}$, time interval between avulsions 50 [yr].





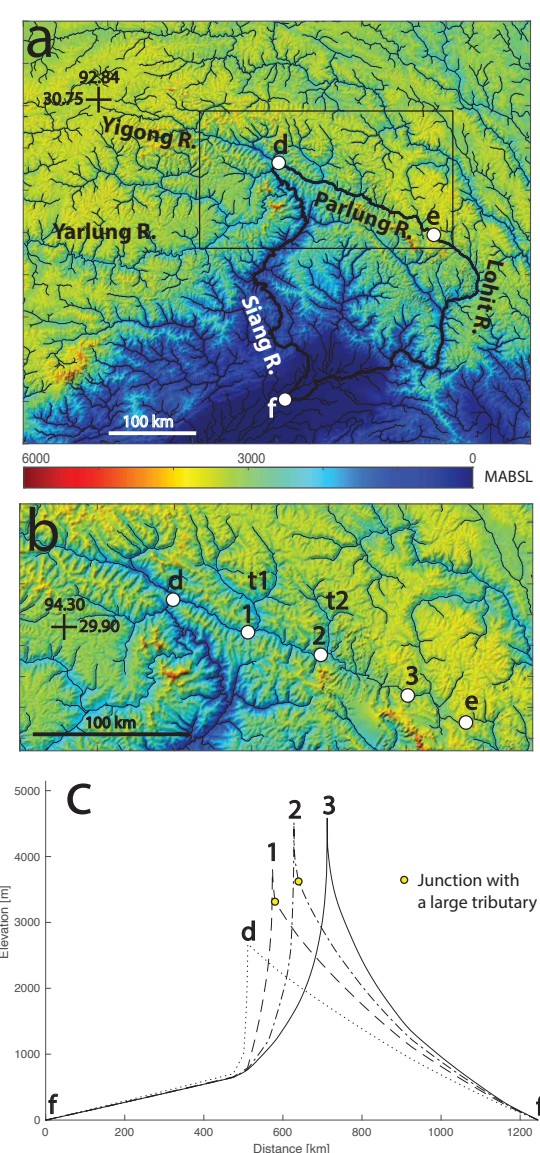

**Figure 6.** Simulated windgap migration along the Parlung valley. (a) A regional map, based on 15 arc-second GMTED2010 DEM (approximately 500 m resolution) showing the Yigong, Parlung, Lohit, Siang and Yarlung rivers (location is shown as a box in figure 2a). Reconstructions of a paleo-drainage pattern (*Lang and Huntington*, 2014; *Govin et al.*, 2018; *Zhang et al.*, 2019) suggest that the Yigong and Parlung used to drain southeast to the Lohit, until the Yigong was captured by the Siang river. This created a windgap at the top of the beheaded Parlung valley, just east of the capture point. The approximate capture location is marked by point d. Point e marks the current location of the windgap between the north-west flowing Parlung and southeast flowing Lohit rivers. Point f marks the confluence of the Lohit and Siang rivers. Thin dark lines mark river systems with drainage area larger than $10^8$ $m^2$, and the bold dark lines mark the river system that is simulated in panel c. A black rectangle marks the area shown in panel b. (b) Map of the Parlung river basin. The Parlung reversed its flow direction following the capture, likely through windgap migration from the capture point (point d) to the current location of the windgap (point e). As the windgap migrated, portions of the valley that used to drain eastward through the Lohit reversed their flow direction. Points 1, 2 and 3 mark simulated stable windgap locations in conjunction with panel c. The labels t1, and t2, mark large tributaries of the Parlung river. (c) Profiles of simulated initial and steady-state topography. The symbol f marks the model boundaries at the location of the Lohit-Siang confluence, as shown in panel a. The symbols d, 1, 2, 3, mark the windgap locations, in conjunction with panel b, for the cases of: d - a profile at the time of capture of the Yigong-Parlung by the Siang (the initial topography of the simulations); 1 - a profile of a simulated stable windgap position that develops after the Yigong-Parlung capture caused eastward windgap migration along the Parlung valley. Note that this stable location is just west of a confluence with a large tributary (t1). The confluence location is shown in panel b (point 1) and marked with a yellow circle on the topographic profile in panel c); 2 - a stable windgap position that developed by simulating an avulsion (i.e., shifting the confluence of tributary t1 to the aggressor's side of the windgap: left of the windgap location in profile 1). Note that this new stable windgap position is just west of a confluence with a large tributary t2 (the confluence location is shown in panel b and marked with a yellow circle on the topographic profile in panel c). 3 - a stable windgap position that is attained through avulsion simulation, where tributaries with drainage area larger than $10^7$ $m^2$ are allowed to avulse. The simulations use the following parameters: $n = 1$, $m = 0.5$ (e.g., *Wang et al.*, 2017; *Yang et al.*, 2018), $\delta x = 500$ m (i.e in accordance with DEM resolution), $D = 2$ $m^2$ $yr^{-1}$, $K = 7.96 \times 10^{-6}$ $yr^{-1}$, $U = 0.005$ m $yr^{-1}$.



## 4    Results

Fixed confluence simulations with synthetic topography show that trunk-tributary confluences affect the velocity of windgap migration. Analysis of the windgap location through time (Figure 5c) shows that the velocity of windgap migration decreases as it approaches a trunk-tributary confluence at the victim's side, and increases as the windgap migrates across a confluence.

At a larger scale, the migration velocity decreases as the windgap migrates further from its initial location and closer to the center of the model domain (Figure 5a-d). Importantly, we observe that in cases where the windgaps do not reach the center of the model domain, it attains a stable position close to a confluence with a tributary that drains to the side of the victim basin.

The same simulations further show that the tributaries drainage area influence the location where the windgap attains a stable position, as well as the mean windgap migration velocity (Figure 7a,b). Figure 7a shows that as the relative drainage area of

tributaries increases, the stable windgap position is farther away from the center of the model domain (i.e., closer to the left side of the model, Figure 5). The figure also shows that this position is generally close to a confluence at the victim's side of the windgap (i.e., 7a). Importantly, a co-linear $\chi - z$ relation is observed for the aggressor and victim basins even when the stable windgap position is not at the center of the model domain (Figure 5a-d).

Figure 7a shows that, everything else being equal, the value of the area exponent $m$ influences the position of stable

windgaps. More specifically, when changing the value of the exponent $m$ from 0.55 to 0.45, windgaps attain a stable position closer to the center of the model domain. The distance between stable windgap positions with $m = 0.55$ and $m = 0.45$ typically corresponds to the distance between successive confluences.

Fixed confluence simulations based on the Parlung-Siang-Lohit setting (Figure 6), aimed to explore the effect of trunk-tributary confluences with a natural topology, show that the windgap along the beheaded Parlung Valley (with $m = 0.5$, as

in the simulations of *Wang et al.* (2017) and *Yang et al.* (2018) in the same area), migrated until attaining a stable position relatively close to the capture point and far from the observed location of the current windgap (i.e., points d and e in Figure 6, respectively). This stable windgap position is close to a trunk-tributary confluence on the victim's side (tributary t1 and point 1 in Figure 6b-c, about 150 km downstream of the observed windgap location). In contrast, in a similar experiment with a lower $m$ value ($m = 0.45$) the windgap continued to migrate across this confluence and stopped approximately 30 km from

the current natural windgap location.

Avulsion simulations show that avulsions influence the windgap migration velocity as well as it's stable position. To compare windgap migration velocity between paired simulations with avulsions, without avulsions, and with a constant drainage area for each node, the mean windgap velocity in all models is computed up to the location where the windgap reaches a stable position in the fixed confluences experiments. Figures 5c and 7b show that in synthetic settings, avulsions increase the mean velocity

of windgap migration, and that this velocity scales with the frequency of avulsions (Figure 7c). For the Parlung-Siang-Lohit setting, we ran a simulation without avulsions until the windgap attained the aforementioned stable position at point 1 (Figure 6b), and then forced an avulsion by shifting the drainage area of this tributary to the aggressor's side of the windgap. This caused the windgap to migrate further east until it attained a stable position next to another large trunk-tributary confluence on the victim's side (point 2, just west of the confluence with tributary t2 in Figure 6b-c). Simulations with randomly occurring





avulsions caused migration across these large trunk-tributary confluences and produce a final stable windgap location at point
3 (Figure 6b-c), approximately 30 km from the current natural windgap location.

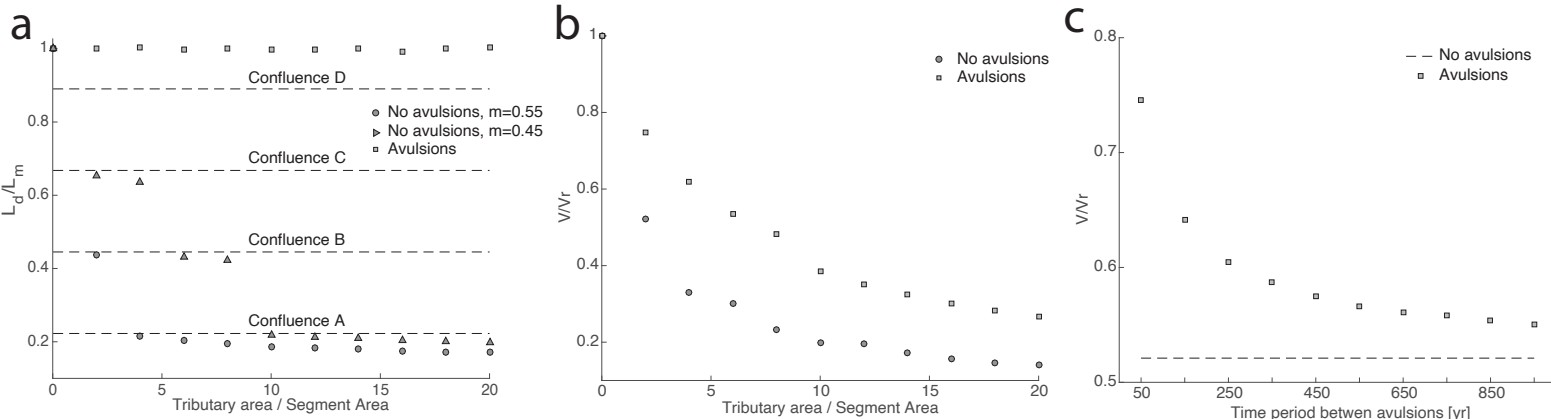

**Figure 7.** The influence of tributary area and avulsion frequency on windgap migration. (a) The influence of tributary area, and the value
of the exponent $m$ on the relative distance between the left boundary of the model domain and the stable windgap position. To account for
tributary spacing and valley width, the drainage area of tributaries is normalized to the area of the valley segment between tributaries (i.e.,
Figure 5a). The windgap migration distance, $L_d$, is normalized by the distance, $L_c$, between the model boundary and the center of the model
domain 5a-b), such that as the windgap migrates closer to this center, the value $L_d/L_c$ is closer to unity. Note that stable windgap positions
are typically next to a confluence on the victim channel whose location is marked by dashed horizontal lines, named A-D in accordance
with Figure 5a. Note that when the trunk-tributary confluences are spatially fixed and the area of tributaries is relatively large, the windgap
can attain a stable position relatively far from the center of the model domain, and that when avulsions are simulated, the windgap attains
a stable position at the center of the model domain, similar to simulation without confluences where all nodes have the same drainage area.
Also note that the distance to stable windgap locations varies with the value of the exponent $m$ (here shown with $m = 0.45$ and $m = 0.55$).
(b) The influence of tributary area (normalized as before) on the mean velocity of windgap migration. This velocity is computed based on
the location and time of where/when the windgap reaches a stable position, and is normalized by the duration it takes for a simulation with
an equal drainage area for all nodes but the same overall model drainage area (i.e., the local drainage area at each node is the mean area
of all nodes for an equivalent model with confluences) to reach the same location. Note that the windgap migration velocity is highest in
simulations with equal drainage area for all nodes, and is also higher in simulations with avulsions compared to those with no avulsions. (c)
The influence of time interval between avulsion on the mean velocity of windgap migration (computed in the same procedure as described
before), for the case where the area of tributaries is twice the segment area. The dashed line marks the velocity of an equivalent simulations
without avulsions. The results plotted in panels a-c rely on a model configuration with 8 trunk-tributary confluences (4 at each side of the
model center), 575 model nodes, $\delta x = 100$ m, $w = 400$ m, $K = 1 \times 10^{-5}$ m$^{1-2m}$yr$^{-1}$, $n = 1$, $m = 0.45$, $D = 0.24$ m$^2$ yr$^{-1}$ (for panel
a also $m = 0.55$, where $D$ is set to $D = 0.68$ m$^2$ yr$^{-1}$ to maintain a constant Pécelt-based length scale with the model of $m = 0.45$ (i.e.,
*Perron et al.*, 2009)).



## 5 Discussion

Fixed confluence simulations indicate that a windgap can attain a stable position that deviates from intuitive expectations. In synthetic simulations, windgaps stabilize away from the center of the model domain, despite the asymmetry in drainage area and slope associated with this position (Figure 5b). Similarly, in the natural topology simulation based on the Parlung-Siang-Lohit system, the simulated windgaps stabilize away from the present-day location of the natural windgap (Figures 5, 6). In these cases, the windgap stabilizes close to a confluence draining to the victim's side of the windgap (Figures 5b, 7a). From a dynamic perspective, this finding suggests that the erosion rate at this proximal confluence is approximately constant and largely independent of the windgap position. This large tributary confluence close to the widngap, counteracts the aforementioned area feedback, by balancing the erosion rates across the windgap, slowing the windgap's migration rate, and in some cases, stopping the migration process entirely and stabilizing the windgap (Figure 5c-d).

From a static perspective, stable windgap positions are possible as long as they conform to a restriction posed by a combination of channel and hillslope relief. In the context of equation 1, and assuming that $D$ and $K$ are spatially uniform and that the spatial transition between a hillslope and a channel is discrete (e.g., *Goren et al.*, 2014a), a windgap is stable as long as the steady-state elevation difference between the two channel heads that bound the windgap is compensated by the hillslope relief:

$$\left(\frac{U}{K}\right)^{1/n}\left|\int_0^{L_{ch1}} A_1(x')^{-m/n}\,dx' - \int_0^{L_{ch2}} A_2(x')^{-m/n}\,dx'\right| \le \frac{U}{2D}\max(L_{h1}^2, L_{h2}^2). \qquad (2)$$

The LHS describes the absolute difference in elevation gain along the channels on both sides of the windgap (subscript 1 and 2) when the morphology of the channels is at geomorphologic steady state and the elevation gain along each channel is integrated from the same elevation. The RHS describes the maxima of the two hillslope reliefs; between the windgap and the channel heads at each of its sides. Note that when the hilslope relief is negligible (i.e., the RHS $\simeq 0$), this condition for a stable divide simplifies to an equality in elevation gain along the two channels (i.e., *Shelef and Hilley*, 2014). The distances between equi-elevation points along channels 1 and 2 (i.e., the location of the lower bounds of integration on the LHS) and the heads of these channels is described by $L_{ch1}$ and $L_{ch2}$, respectively, and the length of the corresponding hillslope is described by $L_{h1}$ and $L_{h2}$. $x'$ represents position along channel and recognizes that drainage area ($A$) varies with this position.

An example of an asymmetric and stable widngap is depicted as case 2 in Figure 5b. The stability of this setting is verified by the co-linear $\chi - z$ relations of the channels that bound the windgap, where $\chi = \int_0^x \left(\frac{A_0}{A_1(x')}\right)^{m/n} dx'$ (e.g., *Perron and Royden*, 2013) (Figure 5d). The co-linearity of the $\chi - z$ profiles indicates that the two channels are at steady state and erode at the same rate. Here, the confluence adjacent to the windgap forms a channel head that is approximately at the same elevation as that on the aggressor's side of the windgap (channel heads were defined by the transition from concave to convex profile). The hillslope relief is larger than the minute difference in elevation between the channel heads, and thus a stable windgap position is attained (see filled circles and bars in the $\chi - z$ plot). Generally, as long as equation 2 is satisfied, the same arrangement of confluences along a valley can produce different stable windgap positions such as those shown in Figures 5b, and 6b-c).



Fixed confluence simulations with synthetic topography (Figures 5, 7a) show that the distance between stable windgap
positions and the center of the model domain increases with the relative drainage area of tributaries (Figures 5b, 7a). From the
static perspective of equation 2, an increase in tributaries' drainage area reduces the elevation gain along the channels and thus
is more likely to facilitate situation where the hillslope relief is larger than the difference in elevation gain between the channel
heads. From a dynamic perspective, and given that the simulation's initial condition is associated with a high asymmetry
in topographic gradient across the windgap (i.e., case 1 in Figures 5b), only confluences with relatively large tributaries are
capable of balancing the shallower gradient along the victim side and ensure equal erosion rates across the widngap (i.e.,
equation 1), stopping its migration closer to the left model boundary. A similar pattern is observed in the simulations of the
Parlung-Siang-Lohit capture, where the windgap stabilizes just before it approaches confluences with large tributaries (points 1
and 2 in Figure 6b). Tributaries of a relatively small drainage area are able to stop the migration process only when the windgap
is closer to the center of the model domain and the overall slope asymmetry across the windgap is relatively small (Figure 7a).
Similarly, a lower value of the exponent $m$ enables windgap migration closer to the center of the model domain because it
decreases the dependency of (a) fluvial erosion (i.e., equation 1), and (b) the elevation gain along steady state channels (i.e.,
equation 2), on the distribution of drainage area along the valley (i.e., *Shelef and Hilley*, 2014).

The velocity of windgap migration in fixed confluence simulations changes as confluences are being traversed, and the mean
velocity decreases as the area of side tributaries increases (Figures 5c, 7b). The decrease in windgap velocity as it approaches
a confluence reflects an increased balance in erosion rate across the windgap that stems from a relative increase in topographic
gradient in the victim's side of the windgap, between the migrating windgap and the erosionally stable confluence. Once the
windgap traverses the confluence, the tributary's discharge shifts to the aggressor basin. This amplifies the erosion rate at the
aggressor's side of the windgap compared to the victim's side, and thus increases the windgap's velocity. The duration of
decreased velocity can be prolonged (or even infinite if the windgap attains a stable position) compared to the duration of
increased velocity (Figure 5c). Therefore, confluences generally decrease the mean migration velocity compared to reference
simulations (i.e., where the local drainage area is everywhere equal). This effect increases with the drainage area of side
tributaries (Figure 7b), which increases the erosional stability of the confluence. Overall, our results suggest that in areas where
windgaps are common, confluences with large side tributaries may be critical in setting the rate of landscape adjustment to
changes.
The velocity and distance of windgap migration are influenced by the occurrence and frequency of avulsions. Stable windgap
positions can be perturbed by avulsions that route discharge from the victim to the aggressor basin, causing an increase in the
aggressor's erosion rate, a decrease in the victim's erosion rate, and further windgap migration (e.g., the migration of the
windgap from point 1 to 2 in Figure 6b-c). The influence of avulsions on migration velocity is reflected in both the mean
(Figures 7c) and instantaneous (Figures 5c) migration velocity, where everything else being equal, higher avulsion frequency
increases the velocity of windgap migration. It is therefore possible, that in settings where windgap migration is common (e.g.,
Figure 2), the rate of landscape adjustment to changes in tectonic and climate depends not only on the drainage area of side
tributaries, but also on the temporal frequency of avulsions in the alluvial fans at the mouth of the tributaries (e.g., Figure 4).



Avulsions in natural alluvial fans typically occur every few to thousands of years (*Field*, 2001; *Pelletier et al.*, 2005; *Fuller*, 2012). This is a relatively short time scale for large shifts in discharge across the divide given that shifts that occur through
classic basin captures, as described by *Bishop* (1995); *Clark et al.* (2004); *Prince et al.* (2011); *Willett et al.* (2014), are rarely observed (e.g., *Fan et al.*, 2018). Overall, basin captures that are triggered by avulsion are likely frequent, and focusing on such settings may provide ample field examples of recent fluvial response to basin capture (e.g., Figure 4). Further, landscape evolution models tend to preserve antecedent patterns and show a relatively minor tendency for reorganization (i.e., *Kwang and Parker*, 2019). It could be that incorporating avulsion dynamics even in detachment limited settings could reveal an important
component that drives models toward more realistic outcomes. Given that the frequency of avulsions depends on the micro-topography of the system, sediment characteristics, and the magnitude, burstiness, and sequencing of floods (*Field*, 2001; *Stock et al.*, 2008; *de Haas et al.*, 2016; *Leenman and Eaton*, 2020), such modeling efforts may reveal new mechanisms through which climate, lithology and tectonics influence the rate of landscape response.

Whereas a synthetic model without avulsions facilitates stable windgap positions that are far from the center of the model do-
main (Figure 7a), simulations show that avulsions trigger further windgap migration towards the center of the model domain, where the windgap attains a stable position despite continued avulsions (Figure 7a). Similarly, it is possible that avulsions helped the Parlung valley windgap to migrate across tributary confluences to its current location (i.e., point e in Figure 6b). Overall, our results demonstrate that although the same arrangement of trunk-tributary confluences along a valley can generate different stable windgap positions (Figures 5, 6, 7), symmetric positions at the center of the model domain are more stable
than others to perturbations caused by avulsions (Figures 7). By analogy to optimal channel networks (OCN) (e.g., *Rinaldo et al.*, 1992; *Sun et al.*, 1994a, b), stable windgap positions away from this favorable location represent local energetic optimum that, once perturbed by avulsions, develop towards a global optimum in which the windgap is close to the center of the model domain. This analogy is supported by tracking the energy dissipation over a fixed-confluence simulation that produces a stable windgap position and then perturbing it by simulating avulsions (Figure 8). From an OCN perspective, the perturbations
introduced by avulsions enable the system to exit a local minima by temporally increasing the energy dissipation of the system, which is analogous to an annealing procedures (*Sun et al.*, 1994a; *Colaiori et al.*, 1997) used in OCN simulations. Thus, avulsions may act as a natural "annealing" mechanism, that shifts the landscape towards stable configurations that are energetically favorable.

Although our findings clearly demonstrate the influence of tributaries and their avulsions on wingap migration, they are
based on a relatively simple set of assumptions and simulations and a limited number of field observations. For example: we use a detachment limited model to simulate channel erosion, which was used before in similar settings (*Yang et al.*, 2020) and is consistent with the incision into bedrock in sites such as the Parlung-Siang-Lohit capture or into cohesive sediments observed in the Negev field sites (e.g., *Harel et al.*, 2019). However, given that alluvial fans are often associated with transport limited conditions (at least periodically (*Spelz et al.*, 2008)), and that valleys are often filled with unconsolidated sediments, it
is likely that a model that combines detachment and transport limited processes will more accurately describe such settings. Similarly, the hillslope processes in our simulations rely on a linear diffusion approach (*Culling*, 1963) and do not account for the potential influence of subsurface flow and landsliding on the migrating windgap (*Brocard et al.*, 2011, 2012). The



simulations further neglect variabilities in the base level elevations, uplift rate, and lithology (i.e., *Harel et al.*, 2019), and they do not account for flow bifurcations that can split a tributary's discharge to multiple confluences. Finally, whereas our one dimensional simulations likely capture the basic dynamics of windgap migration, two dimensional simulations might reveal a more detailed response.

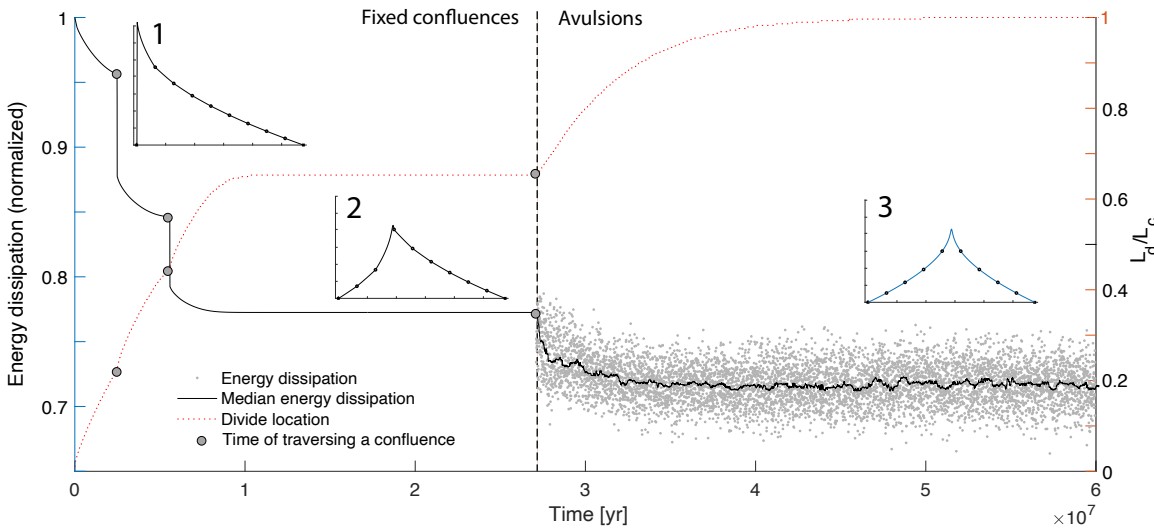

**Figure 8.** The influence of avulsions on energy dissipation and windgap location. Model run time is shown on the x-axis, energy dissipation (normalized by its maximal value) on the left y-axis, and normalized divide location on the right y-axis (i.e., $L_d/L_c$, as in Figure 7). The plot shows the results of a fixed confluence simulation that produced a stable windgap position away from the center of the model domain at approximately $2.7\times10^7$ yr. An avulsion simulation introduced at this point perturbed this stable topography and triggered further divide migration to the center of the model domain. Avulsions can temporally increase the energy dissipation of the system, for example, see the increased energy dissipation value in individual time steps (grey dots) right after the transition to an avulsion simulation. However, eventually they shift the system to a configuration of lower energy dissipation. Note the abrupt decrease in energy dissipation as confluences (grey circles) are being traversed. Inset images 1-3 correspond to the topographic profiles in different stages in this experiment (similar to those in 5b), and show the initial topography (1), a stable asymmetric divide position attained with fixed confluence simulations (2), and a stable symmetric divide position attained with an avulsion simulation (3). These simulations rely on a similar model setting and parameters as in Figure 5. The relative energy dissipation ($P$) was computed following (*Sun et al.*, 1994a, b) as $P \propto \sum_i^N A_i^{1-m/n} \delta x$, where $A_i$ is drainage area of the $i$'th node, and $\delta x$ is the distance between nodes. $N$ is the number of nodes in the simulation, excluding the hillslope nodes close to the windgap (defined by their convex topography). Note that this approach computes the energy content of a steady state topography associated with a given distribution of drainage area.

## 6 Conclusions

In tectonically active and structurally deformed areas, where elongate valleys are common, windgaps can migrate along such valleys and traverse confluences with side-tributaries that drain into the valley. These confluences are stable with respect to

Earth **Surface**
**Dynamics**
Discussions

drainage divide migration, namely, migration does not eradicate them, and can thus influence the migration dynamics by: (1) causing fluctuations in the velocity of windgap migration, where this velocity decreases before the windgap traverses a confluence, and increases right after it traverses the confluence; (2) facilitating multiple configurations of stable windgap locations, aside from the perfectly symmetric configuration, where typically, the windgap stabilizes close to a confluence in the victim channel. The location of these stable configurations is sensitive to the drainage area of side tributaries and the sensitivity

of erosional processes to drainage area (i.e., the exponent $m$ in Equation 1).

    Avulsions of tributaries can abruptly shift discharge across the windgap, and thus change the distribution of erosion across it. Such avulsions can perturb stable windgap positions, and facilitate further windgap migration, where the velocity of windgap migration increases with the frequency of avulsions. From an energetic perspective, such avulsions may be analogous to a natural annealing mechanism, that drives the channel system towards a global energetic optimum.

Overall, our results suggest that tributaries and their avulsions may play a critical role in determining the extent of river basins in tectonically active and/or structurally deformed areas where elongate valleys are common, and thus the partitioning of discharge, erosion and sediments between these basins. Further, the rate of landscape adjustment, even in bedrock dominated mountainous regions, may be modulated by the frequency of such avulsions.

*Author contributions.* ES and LG conceptualized the project and designed the simulations. ES analyzed field examples, developed the code,

run the simulations and analyzed them, wrote the manuscript, and prepared the figures. LG contributed to simulation interpretation and code developement and provided input on the manuscript text and figures.

*Competing interests.* The authors declares that there is no conflict of interest

*Acknowledgements.* This research was supported by Grant No 2019656 from the United States-Israel Binational Science Foundation (BSF) and by Grant No. 1946253 from the United States National Science Foundation (NSF-Geomorphology and Land-use Dynamics). We thank

Onn Cruvi, Elhanan Harel, Tianyue Qu, Philip Prince, and Sean Gallen for valuable discussions. TanDEM-X data were kindly provided by German Aerospace Center (DLR) (project ID: DEM_GEOL1399).





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
