# Peer review of "The rate and extent of windgap migration regulated by tributary confluences and avulsions"

_Earth Surface Dynamics, 2021_

## Author Response (AR1)

We thank Drs. Beeson and Clubb for the constructive and insightful reviews and for putting time and effort into thoroughly reviewing our manuscript. Also, we apologize for the slow response, ES was doing field work and thus we were unable to address comments earlier. Below we address the comments in detail.

Reply to review by Dr. Helen Beeson

**This paper investigates the dynamics of windgap migration using 1-D numerical modeling. In particular, it explores how tributaries and avulsions of those tributaries influence windgap migration rate and stability. The authors present a series of simulations that show that the topology of the network plays a critical role in windgap migration dynamics, influencing the stable location of the windgap, as well as mean windgap migration velocity and how that velocity changes through time (ranging from punctuated to gradual). They show that random processes (avulsion is used here) can trigger a divide that is stable but in a non-optimal location to begin migrating towards a more energetically optimal location. This paper is well written, with a clear hypothesis and experimental design, and it is timely in that it addresses a unique case of divide migration, a subject of much recent interest in the Earth surface process community. I recommend that it be accepted with minor revisions as I have only a few simple questions and a handful of language comments.**

**Is it a given that windgaps migrate or are these channel-head on channel-head windgaps unique? I have seen many of the type of windgaps you show (there are many in the Apennines) and I agree that they clearly migrate, but I'm not sure windgaps formed by lateral capture of headwaters always do. Maybe following capture, a tributary would form and then push the windgap down the main valley of the losing basin (as in your Parlung-Siang-Lohit example), but the basin could also continue to lose area via continued lateral captures. It is clearly out of the scope of this paper to determine in what scenarios windgap migration occurs following capture, but I think it would be good to recognize this question in some way, either in the discussion or in the introduction.**

Thank you for bringing this up. To address this important point we now recognize that the entire basin may lose area by modifying the following sentence in the discussion (L292-295) "Finally, whereas our one dimensional simulations likely capture the basic dynamics of windgap migration along valleys, they do not capture two-dimensional interactions such as drainage area exchange through divide migration along the ridgelines that bound the valley. Two-dimensional simulations might therefore reveal more detailed responses, which could depend on the 2D valley and confluence geometry." We note that side-captures of the valley itself are likely rare when the ridgelines are meaningfully higher than the valley, as in the case of the examples we explored in the Negev desert and Himalaya.

**Similarly, in the last paragraph on page 2 I think an introduction to the idea that windgaps can migrate along valleys could be added and that when they do, side tributaries are preserved.**

Thank you, to clarify this we modified L48-49: "and thus the morphology of the bounding ridge lines and the tributaries that drain them into the valley can be preserved as the windgap migrates along the valley." This is also demonstrated in figure 1.

**Are there processes other than avulsions that might have a similar effect and make this model/idea applicable to regions without alluvial fan-forming tributaries? Could ground water seeping have a similar effect but on a longer time scale? Eventually the area-gaining basin seeps enough ground water to be able to capture another tributary and advance to a more energetically optimal geometry. Another idea is capture of losing basin tributary by gaining basin tributary. I suggest adding a few sentences to the discussion on other potential mechanisms that make this concept more widely applicable (which I think it is).**

Thank you for this suggestion, we agree that other processes, such as seepage and slope failure can further advance this process and now modified L282-285 to acknowledge this "Similarly, the hillslope processes in our simulations rely on a linear diffusion approach (Culling, 1963) and do not account for the potential influence of subsurface flow (e.g., seepage) and landsliding on the migrating windgap (Brocard et al., 2011, 2012)"

We also agree that other processes can cause abrupt changes in discharge across windgaps, and now address this in L289-290 "We also note that processes such as valley damming by landslides or glaciers can cause overflows across windgaps and perturb stable windgap positions".

**Make fig 7 be fig 6 (reorder) – Fig 7 is mentioned before Fig 6.**

Thank you for noticing this, this is now changed.

**Fig 6C It would be nice to see the profiles for current windgap location on here also.**

Thank you for suggesting this - we prefer not to do so as it may be interpreted as if we aim to produce a model that describes this natural topography, whereas our purpose here is to use the topology of the Parlung-Siang system to demonstrate different steady state scenarios and the effect of avulsion on transitions between them. We try to clarify this in lines 147-148 "Note that this simulation aims to demonstrate the potential influence of network topology on windgap migration in a natural setting, and not to investigate the development of the Parlung-Siang-Lohit capture specifically."

**Fig 7 (a) Should y-axis label be Ld/Lc? Legend triangle is tilted compared to those in plot. Maybe state in caption that every marker represents the results of a single simulation?**

Thank you for noticing these mismatches in the plot. They are now fixed. We also changed the caption to clarify that every marker represents the results of a single simulation.

**7b caption needs V and Vr inserted after their explanation (I don't think what they represent is stated elsewhere).**

Thank you for noticing that - we now inserted these symbols.

**All the figure captions are quite long. They could be shortened by removing some of what is already described by legends in the figure. Also, some of them have lengthy interpretations in them that seem like they should go in the main text.**

Thank you, to address this comment, as well as a similar comment by reviewer 2, we shortened the captions of most figures, moved methodological descriptions to the SI, and describe the values of model parameters in a separate table.

**Line comments (mostly typo callouts and language suggestions)**

Thank you for identifying and pointing at these glitches and problems – below we address each:

**4 maybe "in some tectonically active regions…"**

*Thank you - we changed this to: "in some tectonically affected regions…", as not all areas are currently tectonically active.

**4-6 very biased by the study region**

Agreed, we think that changing the previous line to "in some tectonically affected regions…" as you suggested addresses this.

**6 describe the geometry as channel-head on channel-head windgap?**
**Or maybe "Channel-head on channel-head windgap geometry indicates windgap migration with distinct dynamics and potentially quantifiable rates" or something**

We appreciate the value of envisioning the channel head geometry, but we want to emphasize the role of avulsion here, regardless of the specific channel head geometry, and thus prefer to leave the text as is.

**30 maybe "rapidly eroding to the slowly eroding"**

Thank you, we modified accordingly.

**33 change "fast" to "rapidly"**

done

**37 "on the victim basin that loses drainage area"**

done

**39 lengthens**

done

**42 change "whereby" to "in which"**

done

**61 change "set" to "seek"**

done

**83 prominant should be prominent**

done

**115 Clarify that all nodes (both between and with tributaries) are given a local drainage area (if I understand correctly).**

done

**125 "these dynamics"**

done

**130 I suggest saying here at the end that you ran all three versions of the simulation (avulsions, no avulsions m=0.45, no avulsions m=0.55) for ten different values of tributary area/ segment area. It took me a little while to understand that every point on fig 7a was a different simulation.**

Thank you for suggesting this - done

**136 "were" instead of "where"**

done

**171 "its" instead of "it's"**

done

**189 windgap misspelled**

done

**206 windgap misspelled**

done

**220 same**

done

**274 same**

done

Reply to review by Dr. Fiona Clubb

**This is an extremely well-written paper which tackles the interesting problem of windgap migration with a rigorous methodology. Given the interest and recent attention to drainage divide migration in the Earth surface processes community, I think this paper will be of great interest and lead to more studies exploring divide migration in valleys. I have some relatively minor comments, but after these are addressed, I think the paper is very suitable for publication in ESurf. I look forward to seeing the final version published!**

Thank you.

**The avulsion of tributaries at windgaps is a really interesting concept for windgap migration. I was wondering at the ability of this process to occur in vegetated landscapes: the examples shown in Fig 4 all seem to be from arid regions with large alluvial fans where avulsions can happen frequently. What would happen in vegetated regions where the tributary channels may be more fixed in their original course? Would you end up with a windgap in a stable position relatively close the original capture point, as shown in the simulations with no avulsions? While I think simulating this is beyond the scope of the paper, it would be good to see some discussion of the types of landscapes where the fixed confluences vs. avulsions scenarios might be applicable.**

We appreciate this point and its broader implication that the climatic conditions could influence the stochasticity of landscape change and, as a consequence, landscape accessibility to energetically more favorable topologies. To acknowledge the potential vegetation/climate

effect, we added a section to the discussion: (L286-289) "We also did not attempt to explore the influence of vegetation (and by extension climate), which can have competing effects of stabilizing channel banks and reducing the frequency of avulsions (Tal and Paola, 2010), on the one hand, but obstructing flow, and causing aggradation and avulsions (McCarthy et al., 1992;Jones and Schumm, 1999), on the other hand." We think that a broader discussion should be left outside of the current manuscript, which we try to keep relatively focused.

**Following on from this, in agreement with reviewer comment 1, I also think it would help the manuscript to acknowledge in which scenarios wind gaps are likely to migrate and where they may be stationary (either in the introduction or discussion).**

Thank you, a criterion for stability is presented in the discussion (L194-207) section that addresses static settings. We now added the following text to acknowledge additional cases that can perturb stable windgap positions: (L289-290) "We also note that processes such as valley damming by landslides or glaciers can cause overflows across windgaps and perturb stable windgap positions".

**For the landscape evolution modelling, I think you ran all your scenarios with n = 1? I suggest running a sensitivity analysis to test the variation where n is not equal to 1 and there is therefore a non-linear relationship between erosion rates and slope, similar to your tests on the scaling between erosion rate and drainage area. This might have an impact on windgap migration rates if some tributaries are steeper or if there is migration through a shallower part of the main valley.**

Thank you for pointing at this gap. We now include the results of such simulations in an SI. Overall this changes some of the details but the overall pattern remains.

**Would the junction angles at tributary junctions influence the rate of windgap migration? It would be interesting to explore whether, if the junction angles in the victim catchment are larger (more perpendicular to the trunk channel), there is less variation in migration rate across a tributary junction. Perhaps for a future paper!**

Thank you for suggesting this. In the simple 1D perspective presented here the junction angle cannot be explicitly represented. In a more realistic 2D setting, we think that this angle could influence the likelihood of avulsion in the downstream vs upstream valley direction and thus the rate of windgap migration. In natural settings, the junction angle could depend on the relative slope, and potentially, order, of the tributary with respect to the main valley, as well as surface and subsurface hydrology, giving rise to the possibility that additional aspects of network topology control the style of widngap migration. Exploring these issues in detail is beyond the scope of the current manuscript but we now acknowledge its potential role by addition the following sentence (L292-293) "Two-dimensional simulations might therefore

reveal a more detailed response, which could depend on the 2D valley and confluence geometry."

**The results of the modelling seem to show that windgaps like to form a stable position at tributary junctions. Is this borne out by results from real landscapes? The earlier figures in the paper show a lot of detected windgaps across the Himalayas and Appalachians. It seems like it would be relatively simple to detect if these are located at major confluences, which would help to strengthen the conclusions of the modelling by showing a nice correlation with observations.**

Thank you for pointing at that. We think that this may be the case if avulsions, or similar processes that perturb stable windgap configurations (e.g., landslide and glacial valley damming) were not having a meaningful effect on windgap migration. Perhaps future work can explore this in conjunction with the influence of vegetation (in light of the prior comment), by comparing windgap locations (i.e., relative to junctions) between vegetated and non-vegetated settings (i.e., a difference may arise if avulsions are meaningfully less frequent in vegetated areas).

**I find it odd that the simulations with avulsions shows a steadier migration rate of the divide compared to the ones with no avulsions (e.g. Figure 5c). I would expect that, if you have a sudden increase in the discharge to the aggressor basin and a corresponding increase in erosion rate, you should have an increase in migration rate with each avulsion? Is there an explanation for this steady rate of migration in the simulations with avulsions?**

Thank you for pointing at this lack of clarity. We now address this in the discussion (L245-249)." Note that avulsions can effectively reduce or prevent windgap slowdown before large tributaries, reducing the temporal variability in migration velocity. The expression of avulsions in the time-location space of Figure 5c, therefore depends on the frequency of avulsion and the spatial resolution of the simulation."

**Line-by-line and figure comments:**

**All figure captions are quite long and have a lot of methodological detail which would be more suited to the main text. I would prefer having the model parameters as a table for reference rather than in each individual caption.**

Thank you, to address this comment, as well as a similar comment by reviewer 1, we made substantial changes to the captions of most figures, moved methodological descriptions to the SI, and describe the values of model parameters in a separate table.

**Line 37: small typo, should be "loses" rather than "losses"**

Done

**Figure 1: this is a nice figure to illustrate the differences in the tributary network between the gradual divide migration and valley divide scenarios. I think the points explained in the caption would perhaps be clearer if the network and/or divides were coloured by the stream order of the tributaries? This would make it clearer that divide 1 was originally a zeroth order divide in panel c and has become a higher order divide in panel d. It would also highlight the changing stream orders of the networks in panels a and b, compared to c and d where the stream orders should stay the same after divide migration.**

Thank you for this suggestion, we changed the line thickness between channels of different order (to avoid color-related confusion).

**Figure 2: In the text, Fig 2 is referenced as showing the location of windgaps along structurally controlled valleys. Although the figure clearly shows there are a lot of wind gaps in these regions, I found the relationship to the trend of the valleys difficult to see, especially in Fig 2a as it's too zoomed out. This would be more convincing if the location of some wind gaps were shown in relation to the strike of valleys/faults.**

Thank you for pointing at that. To better show the association of windgaps with preexisting structures and antecedent topography we changed the color-scheme of figure 2a, added zoomed in panels to figure 2 (now 2b, 2d) and changed the point size in all panels.

**Figure 3: The thin line showing the initial windgap location is quite difficult to see. Can you make this more obvious?**

Thank you for pointing at that. We replaced the thin line with a bold, yellow line

**Figure 7: what is V/Vr?**

Thank you for pointing at that. We now describe this in the figure caption

**Line 189: windgap misspelled**

Done